# Giant topological magneto-optical effect in noncoplanar antiferromagnet

Y. Okamura [1,2,5] ✉, Y. Hayashi[1,5], N. D. Khanh [1], Y. Tokura [1,3,4], S. Seki [1,2] & Y. Takahashi [1] ✉

Geometrical frustration on triangular lattice is expected to exhibit diverse quantum spin and electronic states endowed with emergent electromagnetic phenomena. The all-in-all-out (AIAO)-type antiferromagnetic spin structure is one such example, possessing the scalar spin chirality that generates giant emergent magnetic field with vanishingly small magnetization. Here, we report on the large spontaneous magneto-optical Kerr effect (MOKE) caused by the AIAO/AOAI state in quasi-two-dimensional triangular-lattice compound $CoNb_3S_6$. Over the entire measured energy region from 55 to 2000 meV, the MOKE is found to be dominated only by the spin chirality. Essential role of momentum-space Berry curvature for both MOKE and dc Hall effect is demonstrated by the spectral analysis of optical Hall conductivity derived from MOKE. The figure of merit of observed topological MOKE, light-polarization rotation angle divided by magnetization, largely exceeds other magnets including time-reversal-symmetry broken antiferromagnet $Mn_3Sn$. Our findings demonstrate the strong light-spin coupling through the spin chirality, paving the way for antiferromagnetic spintronics and future optospintronic devices.

Nontrivial spin ordering provides a fertile ground to explore unconventional electromagnetic properties and functionalities in quantum materials. Recent extensive research on time-reversal-symmetry (TRS) broken antiferromagnets has made the important advance in the field of contemporary materials science[1], potentially overcoming the limitation of spintronics based on ferromagnets. This class of materials is expected to exhibit versatile electromagnetic phenomena allowed by TRS breaking[2–7], such as anomalous Hall effect despite vanishingly small magnetization ($M$), which is promising for the next-generation information medium. However, the TRS-broken antiferromagnets are still very rare and their electromagnetic responses are often rooted in the relativistic spin-orbit coupling similar to that of ferromagnets.

Recently, triangular lattice compounds $CoMe_3S_6$ ($Me$ = Nb, Ta) are revealed to be a new member of TRS-broken antiferromagnets (Fig. 1a), which show the large spontaneous Hall effect despite vanishingly small net $M$[8–12]. The polarized neutron scattering experiments

reveal that this unconventional Hall effect is caused by formation of the all-in-all-out (AIAO)-type antiferromagnetic spin structure on the slightly elongated Co-tetrahedron along the $c$ axis (Fig. 1b)[10,11], which can be viewed as the short wavelength limit of magnetic skyrmion lattice[13]. The noncoplanar spin arrangement hosts the scalar spin chirality defined for three neighboring localized spins, $\chi_{ijk} = S_i \cdot (S_j \times S_k)$ ($S_i$: spin at $i$-th site), which corresponds to the solid angle subtended by those spins. The spin chirality acts as the fictitious magnetic field on the charge carriers moving in the background of noncoplanar spin texture even without spin-orbit coupling, resulting in the topological Hall effect (THE). Therefore, the spontaneous Hall effect in $CoMe_3S_6$ is essentially different from the anomalous Hall effect in the ferromagnets and in the representative TRS-broken antiferromagnet $Mn_3Sn$, which is governed by the spin-orbit coupling[2,4–6,14].

The AIAO spin texture in $CoMe_3S_6$ thus provides the new research arena for the TRS-broken antiferromagnets and attracts much

[1]Department of Applied Physics and Quantum Phase Electronics Centre, University of Tokyo, Tokyo, Japan. [2]Research Centre for Advanced Science and Technology, University of Tokyo, Tokyo, Japan. [3]RIKEN Centre for Emergent Matter Science (CEMS), Wako, Japan. [4]Tokyo College, University of Tokyo, Tokyo, Japan. [5]These authors contributed equally: Y. Okamura, Y. Hayashi. ✉e-mail: okamura@ap.t.u-tokyo.ac.jp; youtarou-takahashi@ap.t.u-tokyo.ac.jp

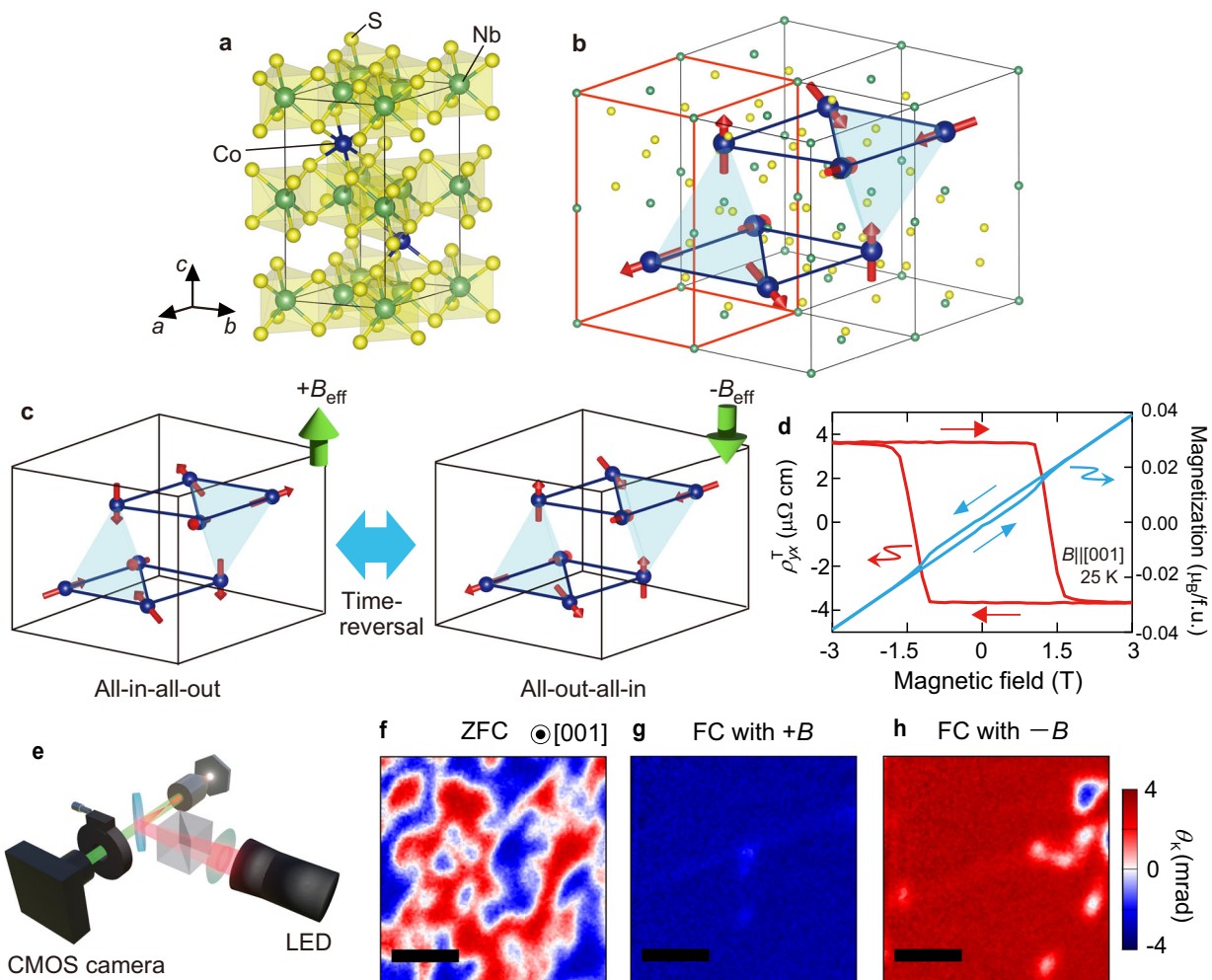

**Fig. 1 | Spontaneous time-reversal symmetry breaking probed by MOKE in AIAO-type antiferromagnet CoNb$_3$S$_6$. a** Crystal structure of CoNb$_3$S$_6$. **b** The magnetic unit cell and schematic illustration of spin structure. The red lines highlight the crystallographic unit cell shown in a. The blue shaded region highlights the tetrahedral Co unit. **c** Schematic illustration of all-in-all-out (AIAO) and all-out-all-in (AOAI) type antiferromagnetic spin structures. This time-reversal pair is characterized by the opposite sign of emergent magnetic field $B_{eff}$ represented by green arrows. **d** Magnetic-field dependence of magnetization $M$ (light blue curve) and topological Hall resistivity $\rho^{T}_{yx}$ after subtracting the normal Hall component (red curve). **e** Schematic illustration of MOKE imaging. **f-h** Magneto-optical Kerr imaging on (001) plane after zero-field cooling (ZFC) (**f**), field cooling (FC) under positive field (**g**), and FC under negative field (**h**). The black scale bar represents the length of 20 μm. These images were taken at zero magnetic field.

attention as the possible long-sought experimental realization of pioneering theoretical predictions[11,15–18]. In the Kondo-lattice model on triangular lattice, the AIAO state can emerge through the Fermi surface nesting, which generates the THE even without magnetic field, as observed experimentally. Notably, in addition to such the topological transport phenomenon, this scalar spin chiral state exhibits the strong coupling to the light through the interband electronic transitions[19–24]. In particular, it is predicted to show the magneto-optical effect induced by the fictitious magnetic field of spin chirality, so-called topological magneto-optical effect[19]. This mechanism does not necessarily require the spin-orbit coupling and net $M$, which is contrasted to the conventional magneto-optical effect in ferromagnets that is proportional to $M$ and governed by the interplay between band exchange splitting and spin-orbit coupling[25,26]. Followed by the prediction, this intriguing optical phenomenon has also been studied in skyrmion systems with finite scalar spin chirality[20,21]; however, the observed response is relatively weak and confined to a limited optical frequency range. We note that the materials proposed in ref. 19, such as γ-Fe$_{1-x}$Mn$_x$ and K$_x$RhO$_2$, are also the important candidates showing the topological MOKE caused by the AIAO structure in addition to Co$Me_3$S$_6$.

In this work, we show the giant topological magneto-optical Kerr effect (MOKE) induced by the scalar spin chirality of AIAO spin structure in CoNb$_3$S$_6$. By using the magneto-optical imaging and broadband spectroscopy, we observe the large spontaneous MOKE, which shows the two-state values corresponding to the sign of fictitious magnetic field from the scalar spin chirality. This demonstrates the direct observation of topological MOKE with negligible conventional $M$-induced MOKE. The normalized topological MOKE by $M$ is particularly large as compared to the conventional mechanisms originating from the spin-orbit coupling. We also discuss the direct connection to the THE that is ascribed to the momentum-space Berry curvature induced by the formation of AIAO spin structure.

## Results

### AIAO-type antiferromagnet CoNb$_3$S$_6$

Our target material CoNb$_3$S$_6$ is a member of intercalated transition metal dichalcogenides. Fig. 1a shows the crystal structure, where magnetic Co ions are inserted into van der Waals gaps of the parent two-dimensional triangular lattice compound 2H-NbS$_2$. The intercalated Co ions form triangular lattice layers, which can be viewed as √3 × √3 superstructures with respect to the NbS$_2$ ones. The overall

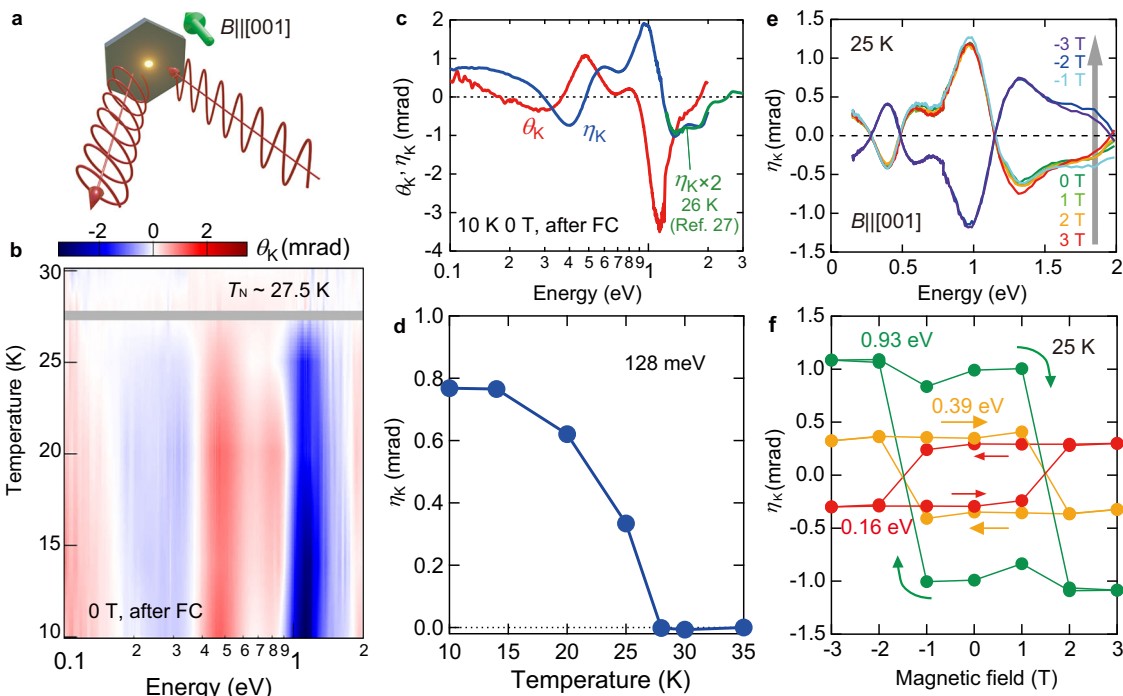

**Fig. 2 | Observation of spontaneous topological MOKE. a** Schematic illustration of MOKE spectroscopy; the magnetic field is applied parallel to *c* axis. **b** Contour map of Kerr rotation angle $\theta_K$ as the function of temperature and photon energy measured for the single domain state stabilized by field cooling at ±1 T from 50 K. **c** Kerr rotation $\theta_K(\omega)$ (red curve) and Kerr ellipticity $\eta_K(\omega)$ (blue curve) spectra at 10 K. The higher-energy $\eta_K(\omega)$ (green curve) is deduced from the magnetic circular dichroism at 26 K reported in ref. 27 The spectral shapes show good agreement with the present result within the overlapping photon-energy range, while the reported spectrum has a smaller magnitude because the measurement temperature is higher than the present study (10 K). **d** Temperature dependence of $\eta_K$ at 128 meV. **e** $\eta_K(\omega)$ spectra when sweeping the magnetic field from 3 T to −3 T at 25 K. **f** Magnetic-field dependence of Kerr ellipticity $\eta_K$ at 0.16 eV (red), 0.39 eV (orange) and 0.93 eV (green) at 25 K.

crystal structure is characterized by the noncentrosymmetric chiral hexagonal space group $P6_322$. The AIAO-type anti-ferromagnetic spin structure is realized at zero magnetic field below the Neel temperature $T_N \sim 27.5$ K, where all spins point outwards in one elongated tetrahedral Co unit and inwards in the other unit, generating the emergent magnetic field along the *c* axis (Fig. 1b, c). The all-out-all-in (AOAI) structure is the time-reversal pair of AIAO state, and these two degenerate domains can be switched by applying magnetic field along the *c* axis (Fig. 1c). The resultant topological Hall resistivity exhibits the large spontaneous component at zero field and rectangular hysteresis in the magnetic-field dependence (Fig. 1d). On the other hand, the net *M* shows the almost linear dependence on the magnetic field with tiny spontaneous component which is less than 0.1 % of the saturation value of $Co^{2+}$ ion (~ 3 $\mu_B$/f.u.) (Fig. 1d).

**Spontaneous topological magneto-optical effect**
We first show the two-dimensional spatial map of MOKE with the 1000 nm-wavelength light incident to the *c* plane (Fig. 1e; see also "Methods"). We clearly observe the multi-domain state exhibiting the positive- or negative-sign optical rotation below the $T_N$ after zero field cooling (Fig. 1f); the magnetic domain whose size is about 20 μm, is of irregular shape, being typical of antiferromagnetic domain free from dipolar interaction. The magnitude of magneto-optical Kerr rotation angle reaches ~ ±4 mrad, which is comparable to the conventional ferromagnets despite the vanishingly small spontaneous *M* in the present material, as discussed later in more detail. The magnetic domains could be poled through the field cooling with magnetic field of ~ ±0.2 T, as shown in Fig. 1g, h. The positive (negative) magnetic field cooling stabilizes the single domain state with negative (positive)-sign Kerr rotation. These observations demonstrate that the MOKE is sensitive to and

directly visualizes the spontaneous TRS breaking of the AIAO antiferromagnet.

To get further insight into the spontaneous MOKE, we next study the broadband spectral response from 0.1 eV to 2 eV, which is measured for the single domain state at zero field after the field cooling (Fig. 2a–c; see also "Methods"). We observe several resonance structures both in Kerr rotation $\theta_K(\omega)$ and Kerr ellipticity $\eta_K(\omega)$ spectra at 10 K (Fig. 2c). Note that these two spectra are connected with each other through the Kramers-Kronig relation. The relatively large Kerr rotation angle of ~ −4 mrad is observed on the resonance around the wavelength of 1000 nm (~ 1.2 eV), being consistent with the imaging measurement (Fig. 1f–h). With increasing the temperature, the magnitude of all the resonances tends to decrease monotonically and disappears above the $T_N$ (Fig. 2b, d).

The observed spontaneous MOKE is caused by the proposed AIAO/AOAI structure rather than the weak spontaneous *M*, which is evident from the magnetic-field dependence (Fig. 2e and f). Figure 2e shows the $\eta_K(\omega)$ spectra when sweeping the magnetic field from +3 T to −3 T. The spectra are unchanged down to −1 T, but the sign change abruptly occurs at −2 T while keeping the overall spectral characteristics and absolute value. This observation suggests that the MOKE shows the two-state value depending on the AIAO or AOAI state at each photon energy. This is further corroborated by the constant energy scan of $\eta_K(\omega)$ as the function of magnetic field (Fig. 2f). We observe the rectangular hysteresis at typical resonance energies without discernible change by applying higher magnetic fields. This field dependence of $\eta_K(\omega)$ obviously shows difference from that of *M*, but is consistent with that of the THE, or equivalently, scalar spin chirality (Fig. 1d). For example, at a photon energy of 0.93 eV, the zero-field magnetization-induced component is roughly estimated to be at most ~ 0.3% of the topological component. Thus, these results demonstrate

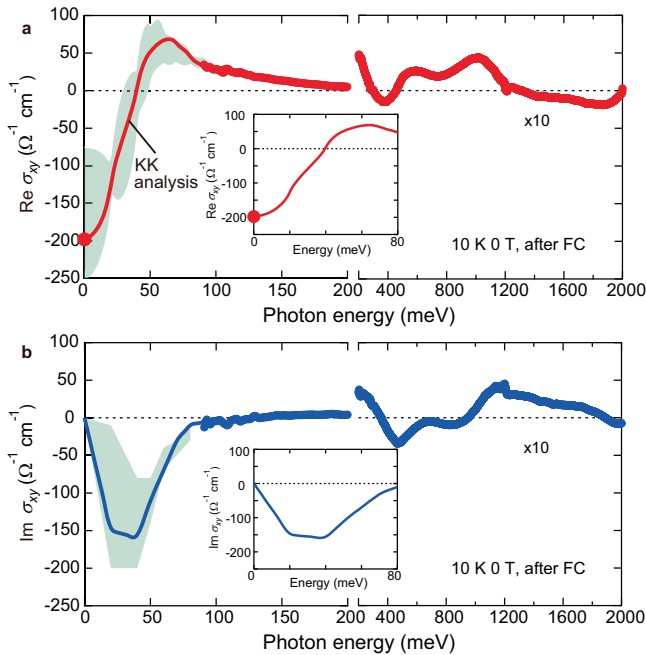

**Fig. 3 | Optical Hall conductivity for spontaneous topological Hall effect.** Real (**a**) and imaginary (**b**) parts of optical Hall conductivity at 10 K, 0 T. The markers above 85 meV indicate the spectra directly obtained by using the experimental $\theta_K$ and $\eta_K$, and curves below 85 meV deduced from the numerical analysis (for more details, see "Methods"). The green shaded region represents the uncertainty resulting from the numerical analysis. The red solid circle in Re $\sigma_{xy}(\omega)$ at zero energy indicates the topological Hall conductivity obtained from the transport measurement. The spectra above 0.2 eV are multiplied by a factor of 10 for clarity. The insets show the magnified view of low-energy resonance around ~ 50 meV.

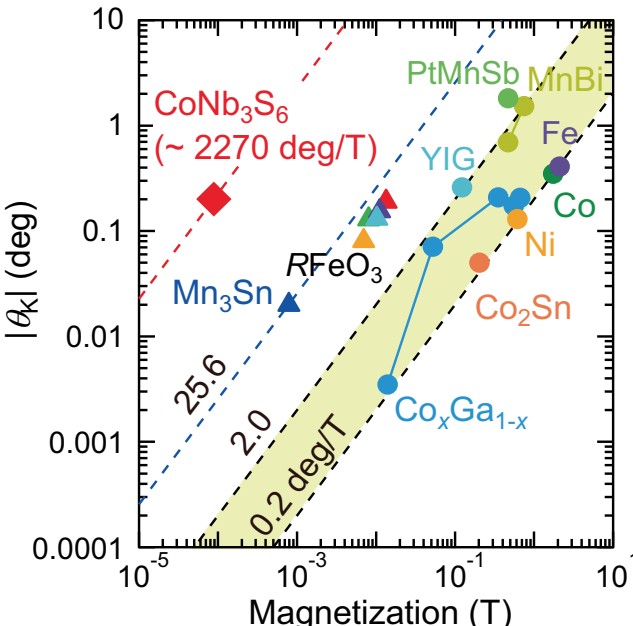

**Fig. 4 | Overview of Kerr rotation angle $\theta_K$ versus magnetization for various magnets.** Circle and triangle symbols represent the data for ferromagnets and antiferromagnets, respectively[6,34–40].

that the observed spontaneous MOKE in all the energy range is dominated by the spin chirality origin, i.e. topological MOKE. The recent MCD spectroscopy reported a similar behavior in a higher-energy region (1.3–3.1 eV, a green curve added in Fig. 2c)[27], showing good agreement with our spectrum. This work also attributed the observed MCD to the AIAO spin structure, as supported by first-principles calculations, being consistent with our discussion.

The observed broadband topological MOKE is located above the energy scale of carrier scattering (≤ 15 meV; see Supplementary Fig. 1), manifesting the impact of fictitious magnetic field, or equivalently scalar spin chirality, on the interband optical transitions. Therefore, the appearance of topological MOKE signals the band reconstruction due to the formation of AIAO spin structure. In addition to the optical responses at finite frequencies, such a band reconstruction is closely related to the THE in the d.c. limit. To pursue this relation, we delve into the optical Hall conductivity $\sigma_{xy}(\omega)$ deduced from the magneto-optical spectra (see also "Methods"). On this basis, we can directly compare the MOKE and d.c. Hall conductivity in the unified unit, and quantitatively evaluate the contribution of band reconstruction to the d.c. Hall conductivity. To be more specific, the integration of Im $\sigma_{xy}(\omega)/\omega$ over frequency, spectral weight, is equal to the d.c. Hall conductivity (Re $\sigma_{xy}(\omega=0)$) according to the sum rule of optical spectra[28–30]. In Fig. 3a, we observe that the real part of $\sigma_{xy}(\omega)$, Re $\sigma_{xy}(\omega)$, shows the notable dispersive spectral shape below 100 meV, while modest structures are discerned above 0.2 eV. The striking feature is that Re $\sigma_{xy}(\omega=0)$ deduced by optical measurement reasonably coincides with the d.c. Hall conductivity obtained by transport measurement (red solid circle in Fig. 3a) within uncertainty of the analysis. Correspondingly, the imaginary part of $\sigma_{xy}(\omega)$, Im $\sigma_{xy}(\omega)$, exhibits the significant peak structure around 50 meV as expected from dispersive

structure in Re $\sigma_{xy}(\omega)$ (Fig. 3b). Thus, the lowest-lying resonance around 50 meV with the largest spectral weight dominates the d.c. topological Hall conductivity.

Since the resonance energy of 50 meV is still larger than the carrier scattering rate, the lowest-lying resonance is caused not by the intraband carrier drifting but by the interband transition on electronic bands near the Fermi level resulting from the band reconstruction due to the AIAO spin structure. Similar to the intrinsic anomalous Hall effect[27–29], the dominant role of interband transitions manifests that the THE is induced by the momentum-space Berry curvature, being consistent with the previous transport measurement and first-principles calculation[12]. Such the momentum-space picture is valid for the small magnetic period limit[31], which is indeed applicable to the present AIAO spin structure. On the other hand, in the skyrmion-hosting B20-type chiral magnets with long magnetic period ranging from a few tens to several hundreds of nanometers, the THE arises from the fictitious magnetic field mainly due to the real-space Berry phase[32,33]. The similar topological MOKE is observed in the skyrmion-hosting system Gd$_2$PdSi$_3$ with a few nanometer magnetic period, while its optical rotation is ~ 10 times smaller and limited to the low-energy region, especially below 1 eV[21]. It is pointed out that Gd$_2$PdSi$_3$ is located in the intermediate region of real-space and momentum-space picture[21,31]. The detailed comparison of two representative materials, CoNb$_3$S$_6$ and Gd$_2$PdSi$_3$, is provided in Supplementary Table 1.

## Discussions

We highlight the figure of merit of the observed topological MOKE by showing the magnitude of Kerr rotation angle with respect to the $M$ (Fig. 4). As discussed in the literature[6], for conventional magnetic systems, there is a general trend that $|\theta_K|$ is proportional to $|M|$. When we introduce $K_S = |\theta_K/M|$ that ranges between 0.2 and 2.0 deg T$^{-1}$ for most of ferromagnets and ferrimagnets[6,34–40], as indicated by the yellow shaded region in Fig. 4. In stark contrast, the topological MOKE in the present system is found to exhibit the gigantic value of ~ 2270 deg T$^{-1}$, which is three orders of magnitude larger than the ferromagnets and almost 100 times larger than the representative TRS-broken antiferromagnet Mn$_3$Sn[6]; we note that the MOKE in Mn$_3$Sn is

dominated by the antiferromagnetic order and that in other reference systems by the magnetization-induced component. This result suggests that the topological MOKE observed here offers clear advantages over the conventional $M$-induced MOKE for ferromagnets and even for $Mn_3Sn$, both of which are governed by the spin-orbit coupling. Namely, the emergent magnetic field in the momentum space stemming from spin chirality exhibits the enhanced MOKE beyond the conventional mechanisms.

The present magneto-optical imaging and broadband spectroscopic measurements unambiguously reveal that the AIAO spin structure shows the topological MOKE in the wide energy region ranging from d.c. to visible energy with no discernible $M$-induced component. This establishes, based solely on experimental results, a direct connection between the observed topological MOKE and the topological Hall effect, and reveals the essential role of momentum-space Berry curvature arising from electronic band reconstruction. We also quantitatively discuss that the topological MOKE in $CoNb_3S_6$ is significantly more pronounced than the magneto-optical response in other materials including the skyrmion system $Gd_2PdSi_3$ and the representative TRS-broken antiferromagnet $Mn_3Sn$ from many viewpoints. In particular, the topological Kerr rotation angle divided by magnetization is quite large as compared to the MOKE governed by the spin-orbit coupling. These results are attributed to the giant emergent magnetic field of AIAO spin structure that can be viewed as the short-wavelength limit of skyrmion lattice with vanishingly small $M$. While the optical writing of the AIAO structure has been already demonstrated via laser-induced heating[27], the strong light-spin coupling inherent to this unique spin structure also suggests the ultrafast control of spin chirality by irradiation of circularly polarized light without laser-induced heating, which has been extensively studied theoretically[22–24]. Thus, the present findings open up a promising pathway to noncontact and fast reading as well as potential writing of antiferromagnetic domains using laser technology towards future optospintronic devices.

## Methods

### Single crystal growth
Powders of elemental Co, Nb and S are mixed together in a silica tube and annealed at 900 °C for 5 days. The process is repeated with regrinding in between, to enhance the quality of the polycrystal. Single crystals of $CoNb_3S_6$ are synthesized by chemical vapor transport (CVT) with iodine as a transport agent. 0.5 g iodine and 2 g polycrystalline $CoNb_3S_6$ powder are loaded into a quartz tube and set into a two-zone furnace for 10 days. The temperatures at the hot and cold zones are kept at 950 °C and 800 °C, respectively. $CoNb_3S_6$ crystallizes in hexagonal thin-plate shape, with the largest surface orthogonal to the $c$-direction. The crystal structure is confirmed by powder X-ray diffraction and Rietveld refinement, and the crystal axes are determined by Laue X-ray back-scattering. The sample used in the present MOKE experiment was taken from the same batch as that used in our previous study on the Nernst effect[12], and therefore the composition is expected to be nearly identical. The exact Co concentration in our $Co_xNb_3S_6$ is estimated to be $x \sim 0.952$ based on the EDX measurement, close to the nominal value $x = 1$.

### Transport and magnetization measurements
The Hall resistivity and magnetization were measured by using Physical Property Measurement System (Quantum Design) and Magnetic Property Measurement System (Quantum Design), respectively.

### Magneto-optical Kerr imaging
Magneto-optical Kerr imaging was performed using a polarizing microscope with nearly crossed polarizer. We used a Halogen lamp as the light source and monochromatized using the bandpass filter with center wavelength of 1000 nm and bandwidth of 50 nm. The linearly polarized near-infrared collimated light was incident to the $c$ plane of sample. The reflected light was guided to a CMOS camera to take the polarization-resolved image of the sample with a nearly crossed analyzer placed before the camera. The weak magnetic field of $\sim \pm 0.2$ T is applied for field cooling using a permanent magnet.

### Broadband magneto-optical Kerr spectroscopy
The polar magneto-optical Kerr spectroscopy was performed with use of a Fourier-transform infrared spectrometer (FTIR) for 0.055–1.2 eV and a monochromator-type spectrometer for 1.2–2 eV. The external magnetic field up to 3 T was applied perpendicular to the sample surface by using a superconducting magnet. For the high-precision polarimetry, we used a photo-elastic modulator (PEM) in conjunction with a MCT detector for 0.1–1.2 eV and a Si photodiode above 1.2 eV[41]. The detection of the fundamental and second harmonic synchronized to the modulation frequency in the reflected light enables us to simultaneously measure Kerr ellipticity $\eta_K$ and Kerr rotation $\theta_K$, respectively. To deduce the Kerr spectra, we anti-symmetrized the spectra for the positive and negative magnetic fields.

For the measurement in 0.055–0.085 eV, we used a liquid He-cooled bolometer detector and put two wire-grid polarizers before and after the sample, which are oriented at 45 degrees with respect to each other. Only the Kerr rotation can be measured from the light intensity at the positive field $I(+B)$ and that at the negative magnetic field $I(-B)$[21,42]:

$$\theta_K = \frac{1}{2} \frac{I(+B) - I(-B)}{I(+B) + I(-B)}. \tag{1}$$

### Optical conductivity $\sigma_{xx}(\omega)$ and optical Hall conductivity $\sigma_{xy}(\omega)$
The optical conductivity spectra were deduced through the Kramers-Kronig transformation of the reflectivity spectra from 0.01 to 4 eV (Supplementary Fig. 1). We measured the reflectivity spectra using a Fourier-transform-type spectrometer in the infrared region (0.01–1 eV) and a mononchromator-type spectrometer in the visible and ultra-violet regions (1–4 eV). For the extrapolation of the reflectivity data, we assumed that the reflectivity is proportional to $\omega^{-4}$ above the highest energy. The optical Hall conductivity spectra above 85 meV were calculated by using the following formula; $\sigma_{xy}(\omega) = -\sigma_{xx}(\omega)\varepsilon_{xx}^{1/2}(\omega)(\theta_K(\omega) + i\eta_K(\omega))$. Below 85 meV, we perform the numerical analysis as discussed in the next section.

We note that linear dichroism has been reported in the sister compound $CoTa_3S_6$[42–44]. However, we can rule out any possible contamination from time-reversal-even linear dichroism in our measurements of the time-reversal-odd MOKE. The Kerr rotation and ellipticity spectra (Fig. 1b, c) are defined as antisymmetrized signals between the AIAO and AOAI states obtained by magnetic-field cooling under opposite field directions. Therefore, the time-reversal even linear dichroism, even if any, can be safely excluded. Furthermore, real-space imaging reveals the presence of only two domains exhibiting polarization rotation with opposite signs (Fig. 1f), which can be controlled by out-of-plane magnetic-field cooling (Fig. 1g, h). The magnitude of the polarization rotation observed in this imaging experiment is in good agreement with that obtained from the Kerr spectra, corroborating that the polarization rotation reported here originates from the MOKE. It should also be emphasized that no evidence of linear dichroism in $CoNb_3S_6$ has been reported in previous studies.

### Numerical analysis of far-infrared $\sigma_{xy}(\omega)$ spectra
Since we measure the Kerr rotation $\theta_K(\omega)$ spectrum from 55 meV to 2 eV and Kerr ellipticity $\eta_K(\omega)$ spectrum from 85 meV to 2 eV (Supplementary Fig. 2), we can deduce $\sigma_{xy}(\omega)$ above 85 meV by direct procedure as discussed above (Supplementary Fig. 3). On the other hand, we deduce the $\sigma_{xy}(\omega)$ spectrum below 85 meV in the following numerical

analysis. When decreasing the energy, Im $\sigma_{xy}(\omega)$ decreases below 0.2 eV with sign change from plus to minus value at ~ 0.12 eV, and Im $\sigma_{xy}(\omega = 0)$ must be zero because of causality constraint (Supplementary Fig. 3). Therefore, at least one negative peak exists in Im $\sigma_{xy}(\omega)$ below 85 meV. Thus, by considering the Im $\sigma_{xy}(\omega)$ values at several discrete energy points, we calculated Re $\sigma_{xy}(\omega)$ spectrum down to zero energy for each assumed negative peak in Im $\sigma_{xy}(\omega)$ through the following Kramers-Kronig relation[42];

$$\text{Re}\,\sigma_{xy}(\omega) = \frac{2}{\pi} P \int_0^\infty \frac{\omega'\text{Im}\sigma_{xy}(\omega')}{\omega'^2 - \omega^2} d\omega'. \qquad (2)$$

By using the calculated Re $\sigma_{xy}(\omega)$ and Im $\sigma_{xy}(\omega)$ spectra, we can analytically calculate the $\theta_K(\omega)$ spectrum down to zero energy. Based on this premise, the presumed negative peak in Im $\sigma_{xy}(\omega)$ is narrowed down to minimize the deviation between the calculated $\theta_K(\omega)$ and the experimental value across the entire energy range (>55 meV). Although we still have some uncertainty as represented by the green shaded regions in Fig. 3, the Re $\sigma_{xy}(\omega = 0)$ deduced from this analysis reasonably coincides with the d.c. Hall conductivity obtained from the transport measurement, validating the present analysis. Among the spectra within the uncertainty, we plot an example of spectrum of Re $\sigma_{xy}(\omega)$ whose zero-energy limit is directly connected to the d.c. Hall conductivity obtained by the transport measurement, which is shown by the red solid curve in Fig. 3a; the corresponding Im $\sigma_{xy}(\omega)$ is shown by the blue solid curve in Fig. 3b. For more details, see also Supplementary Fig. 4.

## Data availability
All experimental data are available at Zenodo at https://doi.org/10.5281/zenodo.18976681.

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

## Acknowledgements

We thank R. Arita, M. Hirschberger and S. Okumura for valuable comments, and K. Shoriki and Y. D. Kato for experimental help. This work was partially supported by JSPS KAKENHI (Grant Nos. 21H04990, 22H04965, 23K13069, 23H05431, 24H02235, 25H00611, 25K00956 and 25K22214), JST grants JPMJFR212X, JPMJCR23O4 and JPMJPR2594, Murata Science Foundation, and Asahi Glass Foundation.

## Author contributions

Y.O. and Y.H. performed the optical experiment and analyzed data under supervision of Y. Takahashi. N.D.K. prepared the sample under supervision of S.S. All authors discussed and interpreted the results with inputs from other authors. Y.O. and Y. Takahashi wrote the manuscript with assistance of other authors. Y. Takahashi, Y. Tokura, and S.S. designed and supervised the project.

## Competing interests

The authors declare no competing interests.
