## [Transparent Peer Review file · Nature Communications]

Giant topological magneto-optical effect in noncoplanar antiferromagnet

Corresponding Author: Professor Yoshihiro Okamura

Version 0:

Reviewer comments:

Reviewer #1

(Remarks to the Author)

The manuscript "Giant topological magneto-optical effect in noncoplanar antiferromagnet" presents a comprehensive experimental study of a giant topological magneto-optical Kerr effect (MOKE) in the AIO-type noncoplanar antiferromagnet CoNb_3S_6 . The Kerr response is shown to persist continuously from the infrared to the visible spectral range and is attributed entirely to scalar spin chirality rather than magnetization-driven mechanisms. The extraction of the optical Hall conductivity further substantiates the topological origin of the effect, representing a significant advancement in antiferromagnetic spintronics and magneto-optical research. The manuscript is generally well organized, the data quality is high, and the findings are potentially impactful. I recommend publication after revision.

Comments for Revision

1. The authors state that the observed MOKE originates solely from scalar spin chirality (Fig.2), but the justification for excluding magnetization-induced contributions could be expanded. A quantitative upper bound estimation of the magnetization-driven component, along with a clearer comparison to Mn_3Sn and other reference systems in Fig.4, would reinforce this conclusion.
2. The derivation of $\sigma_{xy}(\omega)$ below 85 meV relies on numerical Kramers–Kronig inversion. Although the procedure is outlined in Methods, further clarification would improve reproducibility. A brief workflow or pseudo-algorithm, an explanation of how uncertainties in the shaded region of Fig.3 were obtained, and comments on the influence of alternative low-energy peak assumptions on $\sigma_{xy}(0)$ would be helpful.
3. The novelty of this study is clear, but the manuscript would benefit from a more explicit comparison with related reports, including skyrmion-based topological MOKE systems and Gd_2PdSi_3 , particularly regarding physical mechanisms, spectral coverage, and Kerr magnitude. A concise comparison table could help highlight the advantages of CoNb_3S_6 .
4. Following the previous point, the topological MOKE in AIO antiferromagnets was first theoretically proposed in Ref.18. Although conceptually related to skyrmion-induced topological optical effects, the magnetic structures are distinct. Ref.18 should therefore be clearly separated from Refs.19 and 20 when discussing theoretical background. It would also be valuable for the authors to comment on whether similar topological MOKE may be expected in proposed candidate systems such as $\text{Fe}_x\text{Mn}_{1-x}$ or K_xRhO_2 from Ref.18, and under what experimental conditions such signals might be observed.
5. Improving the contrast or adding markers to highlight domain boundaries in Fig.1f–h could aid visualization. Additionally, Fig.3 may benefit from an inset zooming in on the low-energy resonance around ~50 meV.
6. Ref.26 appears not to be cited in the main text. This should be corrected for completeness.

Reviewer #2

(Remarks to the Author)

This paper reports the complex magneto-optical Kerr angle spectroscopy on the unconventional magnetic material

CoNb3S6. It is shown that the observed signal cannot arise from a net magnetization, and it is instead attributed to a tetrahedral AIAO/AOIA AFM structure. The spectroscopic measurements are carefully done and analyzed, span a large energy range and are of high resolution. They are an important contribution to the literature, in particular for providing the microscopic understanding of the MOKE signal.

However, I have a serious concern about the novelty, appropriate citations, and some questions about the interpretation of the results.

1) First, in Kirstein et al (<https://doi.org/10.1103/wh5t-12fn>), which is not cited in the manuscript, MCD spectroscopy and domain imaging on CoNb3S6 were already reported. Okamura et al. provide more comprehensive measurements (photon energy range, real and imaginary part), which should definitely be published. However, Kirstein et al already interpreted the spectrum as arising from noncoplanar tetrahedral AFM order, and backed this up with DFT calculations. The conclusions of the manuscript under review are consistent with this, rather than fundamentally new. Further, the paper ends on the potential of using laser writing to encode AFM domains, but this was also already demonstrated in this compound by Kirstein et al.

2) The results throughout are compared to those on the sister compound CoTa3S6, where the same 3-fold symmetric tetrahedral AFM structure was hypothesized. However, it was recently shown that the magnetic structure in CoTa3S6 is in fact anisotropic, and that this is optically detectable through linear dichroism (arXiv:2507.05486, arXiv:2507.08148, arXiv:2507.12588). If the same was true in CoNb3S6, the measurements presented here would be sensitive to both MOKE and linear dichroism. Can the authors rule out a contribution from the linear dichroism? A further related point: the magnetic behavior in CoTa3S6 is extremely sensitive to exact Co stoichiometry. Is the same true in CoTa3S6, and how is the stoichiometry determined?

3) While I agree with the authors that their evidence that MOKE in CoTa3S6 not caused by magnetization is strong, I do not think they can unambiguously claim that the observed MOKE is caused by the AIAO/AOIA structures. This conclusion relies mainly on neutron scattering measurements which identified AIAO/AOIA structures (Ref 10), but those measurements average over a large sample volume and can mistake multidomain samples with single phase multi-Q structures. I would be more comfortable with phrasing along the lines of "MOKE is consistent with proposed AIAO/AOIA structures", rather than claiming that it arises "solely from scalar spin chirality."

Version 1:

Reviewer comments:

Reviewer #1

(Remarks to the Author)

I have carefully reviewed the revised manuscript and the authors' responses, which satisfactorily address my previous concerns. I recommend the manuscript for publication in its current form.

Reviewer #2

(Remarks to the Author)

I thank the authors for their response. They have addressed most of my concerns (contribution from linear dichroism, sample stoichiometry).

However, I remain unconvinced that the work in the new Ref. 27 (of which I am not an author, as the editors are aware) is appropriately attributed, and I would like to see this corrected before I can support publication.

In its current form, the manuscript reads as though Ref. 27 reported measurements without providing a substantive interpretation. This is not accurate. Ref. 27 proposes essentially the same interpretation of the MCD spectrum as arising from the topological magnetic structure that is put forward in the present manuscript, and it should be credited accordingly. I agree that the present work provides important new measurements extending to lower photon energies, leading to new insights, but it is not the first to offer this interpretation of MCD.

In the response the authors say about ref 27.: "Consequently, the discussion remains largely qualitative, and does not uncover the existence of gigantic response and the underlying physical mechanism strongly associated with the electronic structure in sufficient depth." I don't think this is fair: the data are reported in real units, and compared to a theoretical calculation. The quantitative experimental results of Ref. 27 (e.g. Fig 2b) should be commented on and compared to the quantitative experimental results in the submitted manuscript, in the range of photon energies where they both have measurements (at first sight looks consistent, which is great, and important).

Ref. 27 should also be acknowledged for demonstrating domain control, albeit via a mechanism distinct from the one proposed for future work in this paper (as the authors have emphasized in their response).

1. Reflecting Reviewer #1's comment 1, we discussed the quantitative upper bound estimation of the magnetization-driven component (Line 142, Page 7; Line 194, Page 9).
2. Reflecting Reviewer #1's comment 2, we discussed the details of KK analysis in Supplementary Note 1.
3. Reflecting Reviewer #1's comment 3, we added the comparison of two representative materials showing the topological MOKE in Supplementary Table 1.
4. Reflecting Reviewer #1's comment 4, we added a brief discussion to clarify the experimental prospects for these candidate systems (Line 75, page 4).
5. Reflecting Reviewer #1's comment 5, we changed the color variation of Fig. 1f-1h and revised Fig. 3.
6. Reflecting Reviewer #1's comment 6, we reorganized the reference and cite Ref. 26 in the revised manuscript.
7. Reflecting Reviewer #2's comment 1, we refer to Kirstein *et al.* and emphasize our new findings beyond the previous work (Line 142, Page 7; Line 204, Page 10; Line 216, Page 10).
8. Reflecting Reviewer #2's comment 2, we discussed the possible impact of linear dichroism in Method section.
9. Reflecting Reviewer #2's comment 3, we discussed the actual Co concentration of the present sample in Method section.
10. Reflecting Reviewer #2's comment 4, we softened the claim (Abstract and Line 131, Page 6).

Reply to Reviewer #1's comments

Original report

The manuscript “Giant topological magneto-optical effect in noncoplanar antiferromagnet” presents a comprehensive experimental study of a giant topological magneto-optical Kerr effect (MOKE) in the AIAO-type noncoplanar antiferromagnet CoNb_3S_6 . The Kerr response is shown to persist continuously from the infrared to the visible spectral range and is attributed entirely to scalar spin chirality rather than magnetization-driven mechanisms. The extraction of the optical Hall conductivity further substantiates the topological origin of the effect, representing a significant advancement in antiferromagnetic spintronics and magneto-optical research. The manuscript is generally well organized, the data quality is high, and the findings are potentially impactful. I recommend publication after revision.

Comments for Revision

1. The authors state that the observed MOKE originates solely from scalar spin chirality (Fig.2), but the justification for excluding magnetization-induced contributions could be expanded. A quantitative upper bound estimation of the magnetization-driven component, along with a clearer comparison to Mn_3Sn and other reference systems in Fig.4, would reinforce this conclusion.

2. The derivation of $\sigma_{xy}(\omega)$ below 85 meV relies on numerical Kramers–Kronig inversion. Although the procedure is outlined in Methods, further clarification would improve reproducibility. A brief workflow or pseudo-algorithm, an explanation of how uncertainties in the shaded region of Fig.3 were obtained, and comments on the influence of alternative low-energy peak assumptions on $\sigma_{xy}(0)$ would be helpful.

3. The novelty of this study is clear, but the manuscript would benefit from a more explicit comparison with related reports, including skyrmion-based topological MOKE systems and Gd_2PdSi_3 , particularly regarding physical mechanisms, spectral coverage, and Kerr magnitude. A concise comparison table could help highlight the advantages of CoNb_3S_6 .

4. Following the previous point, the topological MOKE in AIAO antiferromagnets was first theoretically proposed in Ref.18. Although conceptually related to skyrmion-induced

topological optical effects, the magnetic structures are distinct. Ref.18 should therefore be clearly separated from Refs.19 and 20 when discussing theoretical background. It would also be valuable for the authors to comment on whether similar topological MOKE may be expected in proposed candidate systems such as $\text{Fe}_x\text{Mn}_{1-x}$ or K_xRhO_2 from Ref.18, and under what experimental conditions such signals might be observed.

5. Improving the contrast or adding markers to highlight domain boundaries in Fig.1f–h could aid visualization. Additionally, Fig.3 may benefit from an inset zooming in on the low-energy resonance around ~ 50 meV.

6. Ref.26 appears not to be cited in the main text. This should be corrected for completeness.

Our response

We thank Reviewer #1 for spending precious time to review our manuscript and highly appreciating our work.

Comment 1: The authors state that the observed MOKE originates solely from scalar spin chirality (Fig.2), but the justification for excluding magnetization-induced contributions could be expanded. A quantitative upper bound estimation of the magnetization-driven component, along with a clearer comparison to Mn_3Sn and other reference systems in Fig.4, would reinforce this conclusion.

Reply 1: The magnetization-induced contribution is so small that it is likely buried in the experimental uncertainty; thus, even as a rough estimate, the magnetization-induced component at zero field is at most on the order of 0.3 % of the topological component at a photon energy of 0.93 eV, which further corroborates the dominance of the topological MOKE. For comparison, in Mn_3Sn , the magnetization-induced component is likewise very small, which cannot be resolved within the experimental accuracy. The Kerr rotation angle divided by magnetization of the topological MOKE in CoNb_3S_6 is approximately two orders of magnitude larger than that observed in Mn_3Sn , as discussed in the manuscript. In other reference systems, the MOKE is dominated by the magnetization-induced contribution.

In the revised manuscript, we added the following sentences:

(Line 142, Page 7) “For example, at a photon energy of 0.93 eV, the zero-field magnetization-induced component is roughly estimated to be at most ~ 0.3 % of the

topological component.”

(Line 194, Page 9) “In stark contrast, the topological MOKE in the present system is found to exhibit the gigantic value of $\sim 2270 \text{ deg T}^{-1}$, which is three orders of magnitude larger than the ferromagnets and almost 100 times larger than the representative TRS-broken antiferromagnet Mn_3Sn ⁶; we note that the MOKE in Mn_3Sn is dominated by the antiferromagnetic order and that in other reference systems by the magnetization-induced component.”

Comment 2: The derivation of $\sigma_{xy}(\omega)$ below 85 meV relies on numerical Kramers–Kronig inversion. Although the procedure is outlined in Methods, further clarification would improve reproducibility. A brief workflow or pseudo-algorithm, an explanation of how uncertainties in the shaded region of Fig.3 were obtained, and comments on the influence of alternative low-energy peak assumptions on $\sigma_{xy}(0)$ would be helpful.

Reply 2: We summarize the workflow of numerical Kramers–Kronig (KK) analysis in Fig. R1(a). The uncertainty of this KK analysis originates from how well the experimental Kerr spectrum above 55 meV, θ_K^{exp} , is reproduced by the Kerr spectrum calculated from an assumed Hall conductivity, θ_K^{calc} . This agreement is quantified by the deviation Δ^2 , where Δ is experimental noise level around 100 meV and set to be 0.65 mrad (Fig. R1(a)). To be more specific, we illustrate three hypothetical $\sigma_{xy}(\omega)$ spectra in the energy range 0 – 85 meV, shown as red, blue, and green curves in Fig. R1(b). From each assumed $\sigma_{xy}(\omega)$, the corresponding θ_K^{calc} spectrum below 85 meV is obtained, as shown in Fig. R1(c). Among these candidates, we retain the θ_K^{calc} spectra that are consistent with θ_K^{exp} within experimental noise level; in this case, the green curve satisfies the criterion and is therefore adopted.

In the revised manuscript, we discussed the details of KK analysis in Supplementary Note 1 as follows. “We summarize the workflow of numerical KK analysis in Supplementary Fig. 4a. The uncertainty in the KK analysis originates from how well the experimentally measured Kerr spectrum above 55 meV, θ_K^{exp} , is reproduced by the Kerr spectrum calculated from an assumed Hall conductivity, θ_K^{calc} . This agreement is quantified by the deviation Δ^2 , where Δ is experimental noise level around 100 meV and set to be 0.65 mrad (Supplementary Fig. 4a). To be more specific, we illustrate three hypothetical $\sigma_{xy}(\omega)$ spectra in the energy range 0 – 85 meV, shown as red, blue, and green curves in Supplementary Fig. 4b. From each assumed $\sigma_{xy}(\omega)$, the corresponding θ_K^{calc} spectrum above 55 meV is obtained, as shown in Supplementary Fig. 4c. From

these candidates, we can judge the consistency of the θ_K^{calc} spectra with θ_K^{exp} within experimental noise level; in this case, the green curve satisfies the criterion and is therefore adopted.”

We also show Fig. R1 in Supplementary Fig. 4.

Fig. R1| Numerical Kramers Kronig analysis. (a) Workflow of KK analysis. (b) Some examples of assumed $\text{Im } \sigma_{xy}(\omega)$. (c) The experimentally measured θ_K^{exp} (black) and the θ_K^{calc} spectra calculated from assumed $\sigma_{xy}(\omega)$ (red, green and blue).

Comment 3: The novelty of this study is clear, but the manuscript would benefit from a more explicit comparison with related reports, including skyrmion-based topological MOKE systems and Gd_2PdSi_3 , particularly regarding physical mechanisms, spectral coverage, and Kerr magnitude. A concise comparison table could help highlight the advantages of CoNb_3S_6 .

Reply 3: We summarize the characteristics of the topological MOKE in CoNb_3S_6 and Gd_2PdSi_3 in Table R1. Because of the intense Berry flux associated with the short-wavelength noncoplanar spin texture, the topological MOKE in CoNb_3S_6 is significantly more pronounced than that in Gd_2PdSi_3 in four key aspects: the spectral region where the topological MOKE is observed, the magnitude of the topological MOKE, its relative strength compared with the magnetization-induced MOKE, and the magnetic-field range over which it appears.

In the revised manuscript, we added this table in Supplementary Table 1. The figure caption is as follows: “We summarize the characteristics of the topological MOKE in two representative materials, CoNb_3S_6 and Gd_2PdSi_3 . Because of the intense Berry flux associated with the short-wavelength noncoplanar spin texture, the topological MOKE in CoNb_3S_6 is significantly more pronounced than that in Gd_2PdSi_3 in four key aspects: the spectral region where the topological MOKE is observed, the magnitude of the topological MOKE, its relative strength compared with the magnetization-induced MOKE, and the magnetic-field range over which it appears.”

	Spin structure	Spectral region	Topological MOKE	Magnetization-induced MOKE	Magnetic field range
CoNb_3S_6	All-in all-out	Entire measured energy range (up to 2 eV)	~ 3.5 mrad	Too small to be resolved	Entire measured field range (0 – at least 3 T)
Gd_2PdSi_3	Skyrmion	Only low energy (up to 0.8 eV)	~ 0.3 mrad	Comparable to topological MOKE	Narrow range ($\sim 0.5 - 1$ T)

Table R1| Comparison of topological MOKE in CoNb_3S_6 and Gd_2PdSi_3 .

Comment 4: Following the previous point, the topological MOKE in AIAO antiferromagnets was first theoretically proposed in Ref.18. Although conceptually related to skyrmion-induced topological optical effects, the magnetic structures are distinct. Ref.18 should therefore be clearly separated from Refs.19 and 20 when discussing theoretical background. It would also be valuable for the authors to comment on whether similar topological MOKE may be expected in proposed candidate systems such as $\text{Fe}_x\text{Mn}_{1-x}$ or K_xRhO_2 from Ref.18, and under what experimental conditions such signals might be observed.

Reply 4: We thank the referee for this important clarification. Following this comment, we reorganized the theoretical background to clearly separate Ref. 18 from Refs. 19 and 20, and explicitly emphasize the difference between the noncoplanar AIAO spin structure and skyrmion textures. Also, regarding the candidate materials proposed in Ref. 18, such as $\text{Fe}_x\text{Mn}_{1-x}$ and K_xRhO_2 , a similar topological MOKE is in principle expected if a stable AIAO-type noncoplanar spin configuration is realized; however, the magnetic structure has not yet been confirmed experimentally. We have therefore added a brief discussion in the revised manuscript to clarify the experimental prospects for these candidate systems as follows:

(Line 75, page 4): “Followed by the prediction, this intriguing optical phenomenon has also been studied in skyrmion systems with finite scalar spin chirality^{20,21}; however, the observed response is relatively weak and confined to a limited optical frequency range. We note that the materials proposed in Ref. 19, such as $\gamma\text{-Fe}_{1-x}\text{Mn}_x$ and K_xRhO_2 , are also the important candidates showing the topological MOKE caused by the AIAO structure in addition to CoMe_3S_6 .”

Comment 5: Improving the contrast or adding markers to highlight domain boundaries in Fig.1f–h could aid visualization. Additionally, Fig.3 may benefit from an inset zooming in on the low-energy resonance around ~ 50 meV.

Reply 5: Thank you for the important suggestions. Following the comments, we revised figures accordingly, as shown below. The domain boundaries are highlighted by changing the color variation (Fig. R2) and we show the magnified view of low-energy resonance in the inset to Fig. 3 (Fig. R3).

Fig. R2| Revised domain imaging. We changed the color variation.

Fig. R3| Revised Hall conductivity spectrum. We show the magnified view of low-energy resonance.

Comment 6: Ref.26 appears not to be cited in the main text. This should be corrected for completeness.

Reply 6: Thank you for your careful reading. We reorganize the reference and cite Ref. 26 in the revised manuscript.

Reply to Reviewer #2's comments

Original report

This paper reports the complex magneto-optical Kerr angle spectroscopy on the unconventional magnetic material CoNb₃S₆. It is shown that the observed signal cannot arise from a net magnetization, and it is instead attributed to a tetrahedral AIAO/AOIA AFM structure. The spectroscopic measurements are carefully done and analyzed, span a large energy range and are of high resolution. They are an important contribution to the literature, in particular for providing the microscopic understanding of the MOKE signal.

However, I have a serious concern about the novelty, appropriate citations, and some questions about the interpretation of the results.

1) First, in Kirstein et al (<https://doi.org/10.1103/wh5t-12fn>), which is not cited in the manuscript, MCD spectroscopy and domain imaging on CoNb₃S₆ were already reported. Okamura et al. provide more comprehensive measurements (photon energy range, real and imaginary part), which should definitely be published. However, Kirstein et al already interpreted the spectrum as arising from noncoplanar tetrahedral AFM order, and backed this up with DFT calculations. The conclusions of the manuscript under review are consistent with this, rather than fundamentally new. Further, the paper ends on the potential of using laser writing to encode AFM domains, but this was also already demonstrated in this compound by Kirstein et al.

2) The results throughout are compared to those on the sister compound CoTa₃S₆, where the same 3-fold symmetric tetrahedral AFM structure was hypothesized. However, it was recently shown that the magnetic structure in CoTa₃S₆ is in fact anisotropic, and that this is optically detectable through linear dichroism (arXiv:2507.05486, arXiv:2507.08148, arXiv :2507.12588). If the same was true in CoNb₃S₆, the measurements presented here would be sensitive to both MOKE and linear dichroism. Can the authors rule out a contribution from the linear dichroism? A further related point: the magnetic behavior in CoTa₃S₆ is extremely sensitive to exact Co stoichiometry. Is the same true in CoTa₃S₆, and how is the stoichiometry determined?

3) While I agree with the authors that their evidence that MOKE in CoTa₃S₆ not caused by magnetization is strong, I do not think they can unambiguously claim that the observed MOKE is caused by the AIAO/AOIA structures. This conclusion relies mainly on neutron scattering measurements which identified AIAO/AOIA structures (Ref 10), but those measurements average over a large sample volume and can mistake multidomain samples with single phase multi-Q structures. I would be more comfortable with phrasing along the lines of “MOKE is consistent with proposed AIAO/AOIA structures”, rather than claiming that it arises “solely from scalar spin chirality.”

Our response

We thank Reviewer #2 for spending precious time to review our manuscript.

Comment 1: First, in Kirstein et al (<https://doi.org/10.1103/wh5t-12fn>), which is not cited in the manuscript, MCD spectroscopy and domain imaging on CoNb₃S₆ were already reported. Okamura et al. provide more comprehensive measurements (photon energy range, real and imaginary part), which should definitely be published. However, Kirstein et al already interpreted the spectrum as arising from noncoplanar tetrahedral AFM order, and backed this up with DFT calculations. The conclusions of the manuscript under review are consistent with this, rather than fundamentally new. Further, the paper ends on the potential of using laser writing to encode AFM domains, but this was also already demonstrated in this compound by Kirstein et al.

Reply 1: Kirstein *et al.* indeed reported a MOKE measurement quite recently, but their study is restricted only to higher-energy window using a conventional technique and lacks quantitative analysis. Consequently, the discussion remains largely qualitative, and does not uncover the existence of gigantic response and the underlying physical mechanism

strongly associated with the electronic structure in sufficient depth. By contrast, the present work substantially advances beyond the previous study by exploiting a newly developed ultrabroadband spectroscopic technique, which enables much deeper insight into the topological transport and optical responses of a noncoplanar antiferromagnet. Below, we summarize the new findings of the present work.

(i) We demonstrate for the first time a gigantic topological MOKE response characteristic of the short-wavelength limit of a skyrmion lattice with vanishingly small net magnetization M , which constitutes the central result of this study. Our ultrabroadband spectroscopy reveals a resonantly enhanced topological MOKE around 1 eV. Notably, the figure of merit of topological MOKE reaches an unprecedented value of approximately 2270 deg/T, exceeding that of conventional ferromagnets by three orders of magnitude and that of the representative time-reversal-symmetry–broken antiferromagnet Mn_3Sn by nearly two orders of magnitude. We further compare our results with the conventional skyrmion system Gd_2PdSi_3 , thereby clarifying the unique characteristics of the short-wavelength noncoplanar spin texture in the present system. These newly discovered quantitative aspects and their physical implications are entirely absent in the previous work.

(ii) By combining ultrabroadband spectroscopy with a newly developed numerical Kramers–Kronig analysis, we show that the lowest-lying resonance in σ_{xy} spectra around ~ 50 meV, which is caused by the electronic bands near Fermi level, plays the dominant role for the topological Hall effect. This finding establishes a direct connection between the observed topological MOKE and the topological Hall effect, and reveals the essential role of the momentum-space Berry curvature generated by the reconstruction of electronic bands. Importantly, this is demonstrated purely from experimental observations. In contrast, although the previous study partially investigates the microscopic origin based on first-principles calculations incorporating the AIAO structure, the analysis remains largely qualitative as the MCD was evaluated only in arbitrary units and within a limited high-energy region above 1.2 eV. Furthermore, the qualitative agreement of spectral characteristics between experiment and calculations is not sufficiently justified. Thus, it is not possible to discuss its relationship to the topological Hall effect, electronic structure and the quantum-geometric Berry curvature. Consequently, the previous work does not address the underlying microscopic mechanism topological physics sufficiently.

We note that our proposed optical writing scheme is based on the inverse effect of the enhanced topological MOKE, which is sensitive to the helicity of light. This is fundamentally different from the mechanism reported by Kirstein *et al.*, where the observed effect is irrespective of light polarization and originates from laser-induced

heating.

In the revised manuscript, we refer to Kirstein *et al.* and emphasize our new findings beyond this work. We revised as follows:

(Line 142, Page 7): “This field dependence of $\eta_K(\omega)$ obviously shows difference from that of M , but is consistent with that of the THE, or equivalently, scalar spin chirality (Fig. 1d). For example, at a photon energy of 0.93 eV, the zero-field magnetization-induced component is roughly estimated to be at most $\sim 0.3\%$ of the topological component. Thus, these results demonstrate that the observed spontaneous MOKE in all the energy range is dominated by the spin chirality origin, i.e. topological MOKE. The recent MCD spectroscopy reported similar behavior in a higher-energy region (1.3 – 3.1 eV)²⁷.”

(Line 204, Page 10): “This establishes, based solely on experimental results, a direct connection between the observed topological MOKE and the topological Hall effect, and reveals the essential role of momentum-space Berry curvature arising from electronic band reconstruction. We also quantitatively discuss that the topological MOKE in CoNb₃S₆ is significantly more pronounced than the magneto-optical response in other materials including the skyrmion system Gd₂PdSi₃ and the representative TRS-broken antiferromagnet Mn₃Sn from many viewpoints. In particular, the topological Kerr rotation angle divided by magnetization is quite large as compared to the MOKE governed by the spin-orbit coupling.”

(Line 216, Page 10): Such strong light-spin coupling inherent to this unique spin structure also suggests the ultrafast control of spin chirality by irradiation of circularly polarized light **without laser-induced heating**, which has been extensively studied theoretically²²⁻²⁴.

Comment 2: The results throughout are compared to those on the sister compound CoTa₃S₆, where the same 3-fold symmetric tetrahedral AFM structure was hypothesized. However, it was recently shown that the magnetic structure in CoTa₃S₆ is in fact anisotropic, and that this is optically detectable through linear dichroism (arXiv:2507.05486, arXiv:2507.08148, arXiv :2507.12588). If the same was true in CoNb₃S₆, the measurements presented here would be sensitive to both MOKE and linear dichroism. Can the authors rule out a contribution from the linear dichroism?

Reply 2: We can rule out any possible contamination from time-reversal-even linear dichroism in our measurements of the time-reversal-odd MOKE. The Kerr rotation and ellipticity spectra (Figs. 1b and 1c) are defined as antisymmetrized signals between the AIAO and AOAI states obtained by magnetic-field cooling under opposite field directions.

Therefore, the time-reversal even linear dichroism, even if any, can be safely excluded. Furthermore, real-space imaging reveals the presence of only two domains exhibiting polarization rotation with opposite signs (Fig. 1f), which can be controlled by out-of-plane magnetic-field cooling (Figs. 1g and 1h). The magnitude of the polarization rotation observed in this imaging experiment is in good agreement with that obtained from the Kerr spectra, corroborating that the polarization rotation reported here originates from the MOKE. It should also be emphasized that no evidence of linear dichroism in CoNb_3S_6 has been reported in previous studies.

In the revised manuscript, we added this discussion to Method section. “We note that linear dichroism has been reported in the sister compound CoTa_3S_6 ^{42,43}. However, we can rule out any possible contamination from time-reversal-even linear dichroism in our measurements of the time-reversal-odd MOKE. The Kerr rotation and ellipticity spectra (Figs. 1b and 1c) are defined as antisymmetrized signals between the AIAO and AOAI states obtained by magnetic-field cooling under opposite field directions. Therefore, the time-reversal even linear dichroism, even if any, can be safely excluded. Furthermore, real-space imaging reveals the presence of only two domains exhibiting polarization rotation with opposite signs (Fig. 1f), which can be controlled by out-of-plane magnetic-field cooling (Figs. 1g and 1h). The magnitude of the polarization rotation observed in this imaging experiment is in good agreement with that obtained from the Kerr spectra, corroborating that the polarization rotation reported here originates from the MOKE. It should also be emphasized that no evidence of linear dichroism in CoNb_3S_6 has been reported in previous studies.”

Comment 3: A further related point: the magnetic behavior in CoTa_3S_6 is extremely sensitive to exact Co stoichiometry. Is the same true in CoTa_3S_6 , and how is the stoichiometry determined?

Reply 3: The sample used in the present MOKE experiment was taken from the same batch as that used in our previous study on the Nernst effect (N. D. Khanh *et al.*, *Nat. Commun.* **16**, 2654 (2025)), and therefore the composition is expected to be nearly identical. The exact Co concentration in our $\text{Co}_x\text{Nb}_3\text{S}_6$ is estimated to be $x \sim 0.952$ based on the EDX measurement, close to the nominal value $x = 1$ (Fig. R4). In fact, the samples exhibit clear hysteresis in magnetization as well as a spontaneous Hall resistivity of approximately $4 \mu\Omega \text{ cm}$, indicating that the composition is close to the ideal value. In contrast, when the deviation from $x = 1$ becomes significant (e.g., $x \sim 0.9, 1.05$), both the

spontaneous Hall resistivity and magnetization are strongly suppressed, suggesting that the magnetic structure is altered, as reported in *Phys. Rev. B* **103**, 184408 (2021).

We added this fact in Method section as follows. “The sample used in the present MOKE experiment was taken from the same batch as that used in our previous study on the Nernst effect ¹², and therefore the composition is expected to be nearly identical. The exact Co concentration in our $\text{Co}_x\text{Nb}_3\text{S}_6$ is estimated to be $x \sim 0.952$ based on the EDX measurement, close to the nominal value $x = 1$.”

Fig. R4| EDX mapping taken from N.D. Khanh *et al.* *Nat. Commun.* **16**, 2654 (2025). The experimental Co composition is estimated to be ~ 0.952 , close to the nominal ratio $x = 1$. There is also no evidence for inhomogeneous Co composition gradients.

Comment 4: While I agree with the authors that their evidence that MOKE in CoTa_3S_6 not caused by magnetization is strong, I do not think they can unambiguously claim that the observed MOKE is caused by the AIAO/AOIA structures. This conclusion relies mainly on neutron scattering measurements which identified AIAO/AOIA structures (Ref 10), but those measurements average over a large sample volume and can mistake multidomain samples with single phase multi-Q structures. I would be more comfortable with phrasing along the lines of “MOKE is consistent with proposed AIAO/AOIA structures”, rather than claiming that it arises “solely from scalar spin chirality.”

Reply 4: The neutron scattering experiment in Ref. 10 identifies the magnetic modulation vector $q = (1/2, 0, 0)$, which alone does not uniquely determine the magnetic structure, as

the reviewer is concerned. However, the representation analysis demonstrates that the AIAO/AOIA magnetic structure, which naturally hosts a finite scalar spin chirality, gives rise to the spontaneous topological Hall effect without net magnetization. Taken together, these considerations uniquely identify the AIAO/AOIA structure as the origin of the observed Hall response. The same conclusion has been independently reached by another group, providing additional support for this interpretation.

While we believe these discussions will resolve the magnetic structure at this stage, there indeed remain some issues such as the slight distortion of AIAO structure as indicated by the linear dichroism measurement in the Ta compound. Therefore, following the reviewer's comment, we have revised the manuscript to soften the wording accordingly. In the revised text, we added the following sentences:

(Abstract): “Here, we report on the large spontaneous magneto-optical Kerr effect (MOKE) caused by the AIAO/AOAI state in quasi-two-dimensional triangular-lattice compound CoNb_3S_6 .”

(Line 131, page 6): “The observed spontaneous MOKE is caused by the proposed AIAO/AOAI structure rather than the weak spontaneous M ...”

1. Reflecting Reviewer #2's comment 1, we acknowledged that Ref. 27 discussed the interpretation of the MCD spectrum (Line 148, Page 7), and showed the comparison with the previous work in Fig. 2c.
2. Reflecting Reviewer #2's comment 2, we commented on the optical control of magnetic domains and its microscopic mechanism (Line 218, Page 10).

Reply to Reviewer #1's comments

Original report

I have carefully reviewed the revised manuscript and the authors' responses, which satisfactorily address my previous concerns. I recommend the manuscript for publication in its current form.

Our response

We thank Reviewer #1 for spending precious time to review our manuscript and recommending the publication.

Reply to Reviewer #2's comments

Original report

I thank the authors for their response. They have addressed most of my concerns (contribution from linear dichroism, sample stoichiometry).

However, I remain unconvinced that the work in the new Ref. 27 (of which I am not an author, as the editors are aware) is appropriately attributed, and I would like to see this corrected before I can support publication.

In its current form, the manuscript reads as though Ref. 27 reported measurements without providing a substantive interpretation. This is not accurate. Ref. 27 proposes essentially the same interpretation of the MCD spectrum as arising from the topological magnetic structure that is put forward in the present manuscript, and it should be credited accordingly. I agree that the present work provides important new measurements extending to lower photon energies, leading to new insights, but it is not the first to offer this interpretation of MCD.

In the response the authors say about ref 27.: "Consequently, the discussion remains largely qualitative, and does not uncover the existence of gigantic response and the underlying physical mechanism strongly associated with the electronic structure in

sufficient depth.” I don’t think this is fair: the data are reported in real units, and compared to a theoretical calculation. The quantitative experimental results of Ref. 27 (e.g. Fig 2b) should be commented on and compared to the quantitative experimental results in the submitted manuscript, in the range of photon energies where they both have measurements (at first sight looks consistent, which is great, and important).

Ref. 27 should also be acknowledged for demonstrating domain control, albeit via a mechanism distinct from the one proposed for future work in this paper (as the authors have emphasized in their response).

Our response

We thank Reviewer #2 for spending precious time to review our manuscript and giving us the valuable comment.

Comment 1: In its current form, the manuscript reads as though Ref. 27 reported measurements without providing a substantive interpretation. This is not accurate. Ref. 27 proposes essentially the same interpretation of the MCD spectrum as arising from the topological magnetic structure that is put forward in the present manuscript, and it should be credited accordingly. I agree that the present work provides important new measurements extending to lower photon energies, leading to new insights, but it is not the first to offer this interpretation of MCD.

In the response the authors say about ref 27.: “Consequently, the discussion remains largely qualitative, and does not uncover the existence of gigantic response and the underlying physical mechanism strongly associated with the electronic structure in sufficient depth.” I don’t think this is fair: the data are reported in real units, and compared to a theoretical calculation. The quantitative experimental results of Ref. 27 (e.g. Fig 2b) should be commented on and compared to the quantitative experimental results in the submitted manuscript, in the range of photon energies where they both have measurements (at first sight looks consistent, which is great, and important).

Reply 1: We thank the reviewer for this important comment and for pointing out the need for clearer attribution of Ref. 27. We agree that Ref. 27 not only reported MCD measurements but also discussed the interpretation of the MCD spectrum in relation to the underlying noncoplanar spin structure, as supported by first-principles calculations, which is consistent with our results. In the revised manuscript, we directly refer the MCD

spectrum from Ref. 27 in Fig. 2c (see also Fig. R1). The spectral shape of Ref. 27 (green curve) qualitatively shows good agreement with ours within the overlapping photon-energy range. The difference in the overall magnitude can be attributed to the difference in measurement temperature (10 K in the present case and 26 K for Ref. 27).

Figure R1: Kerr rotation $\theta_K(\omega)$ (red curve) and Kerr ellipticity $\eta_K(\omega)$ (blue curve) spectra at 10 K measured in the present work. The higher-energy $\eta_K(\omega)$ spectrum indicated in green is calculated from the magnetic circular dichroism spectrum at 26 K reported in Ref. 27.

Also, following the reviewer’s suggestion, we now explicitly acknowledge that Ref. 27 discussed the interpretation of the MCD spectrum in terms of the topological magnetic structure in the revised manuscript as follows:

(Line 148, Page 7) “The recent MCD spectroscopy reported a similar behavior in a higher-energy region (1.3–3.1 eV, a green curve added in Fig. 2c)²⁷, showing good agreement with our spectrum. This work attributed the observed MCD to the AIAO spin structure as supported by first-principles calculations, being consistent with our discussion.”

In addition, we added Fig. R1 to Fig. 2c to present the comparison of the Kerr spectra. The corresponding figure caption is revised as follows:

(Line 455, Page 22) “The higher-energy $\eta_K(\omega)$ (green curve) is deduced from the magnetic circular dichroism at 26 K reported in Ref. 27. The spectral shapes show good agreement within the overlapping photon-energy range, while the reported spectrum has a smaller magnitude because the measurement temperature is higher than the present study (10 K).”

Comment 2: Ref. 27 should also be acknowledged for demonstrating domain control, albeit via a mechanism distinct from the one proposed for future work in this paper (as

the authors have emphasized in their response).

Reply 2: We thank the reviewer for this helpful suggestion. In the revised version, we have added a statement recognizing that Ref. 27 reported domain control, while clarifying that the mechanism involved is distinct from the one proposed for future studies in the present work. The corresponding text is as follows.

(Line 218, Page 10) “While the optical writing of the AIAO structure has been already demonstrated via laser-induced heating²⁷, the strong light-spin coupling inherent to this unique spin structure also suggests the ultrafast control of spin chirality by irradiation of circularly polarized light without laser-induced heating, which has been extensively studied theoretically²²⁻²⁴.”